# Agreement between the Open Barbell and Tendo Linear Position Transducers for Monitoring Barbell Velocity during Resistance Exercise

**DOI:** 10.3390/sports7050125

**Published:** 2019-05-23

**Authors:** Adam M. Gonzalez, Gerald T. Mangine, Robert W. Spitz, Jamie J. Ghigiarelli, Katie M. Sell

**Affiliations:** 1Department of Health Professions, Hofstra University, Hempstead, NY 11549, USA; robspitz2@gmail.com (R.W.S.); jamie.ghigiarelli@hofstra.edu (J.J.G.); katie.sell@hofstra.edu (K.M.S.); 2Department of Exercise Science and Sport Management, Kennesaw State University, Kennesaw, GA 30144 USA; gmangine@kennesaw.edu

**Keywords:** velocity-based training, bar speed, movement velocity, weightlifting, squat

## Abstract

To determine the agreement between the Open Barbell (OB) and Tendo weightlifting analyzer (TWA) for measuring barbell velocity, eleven men (19.4 ± 1.0 y) performed one set of 2–3 repetitions at four sub-maximal percentage loads, [i.e., 30, 50, 70, and 90% one-repetition maximum (1RM)] in the back (BS) and front squat (FS) exercises. During each repetition, peak and mean barbell velocity were recorded by OB and TWA devices, and the average of the 2–3 repetitions was used for analyses. Although the repeated measures analysis of variance revealed significantly (*p* ≤ 0.005) greater peak and mean velocity scores from OB across all intensities, high intraclass correlation coefficients (ICC_2,K_ = 0.790–0.998), low standard error of measurement (SEM_2,K_ = 0.040–0.119 m·s^−1^), and coefficients of variation (CV = 2–4%) suggested consistency between devices. Positive (*r* = 0.491–0.949) Pearson correlations between averages and differences (between devices) in peak velocity, as well as associated Bland-Altman plots, showed greater differences occurred as the velocity increased, particularly at low-moderate intensity loads. OB consistently provides greater barbell velocity scores compared to TWA, and the differences between devices were more apparent as the peak velocity increased with low-to-moderate loads. Strength coaches and athletes may find better agreement between devices if the mean velocity scores are only considered.

## 1. Introduction

Velocity-based training has been proposed as a promising methodology to design resistance training programs, and barbell velocity assessment is becoming more popular and widely accepted as an alternative to using percentages of an individual’s one-repetition maximum (1RM) to derive and quantify training loads [1]. This approach is based on the load-velocity relationship, whereby higher loads are moved at slower velocities [2]. Several studies have suggested that the barbell velocity attained during a resistance exercise can be used to determine the relative intensity, (i.e., percentage of maximum) of the load lifted [3,4,5]. Velocity-based training may serve as an auto-regulatory method for identifying optimal training loads while also accounting for fluctuations in muscle performance [1].

Proper implementation of velocity-based training is predicated on the accuracy of the technology used to monitor barbell velocity. Several devices, such as linear position transducers, video systems, and accelerometers, have been developed to measure barbell velocity in resistance training and research settings [6,7,8,9]. Linear position transducers are among the most popular devices because of their accuracy and relative ease of use, allowing for an objective and instantaneous measurement of barbell velocity for the athlete and/or coach [10]. Linear position transducers consist of a sensor with a cable that is attached to the barbell, and measure barbell velocity via cable displacement with respect to time [9]. For over a decade, Tendo^TM^ weightlifting analyzers (TWA) have been a popular brand of linear position transducers in the strength and conditioning field. Previous research in TWA has demonstrated high validity and reliability for mean and peak velocity assessment during resistance exercise when compared to a valid criterion measure, (e.g., Vicon 3D motion capture system) [7]. Unfortunately, the cost of TWA (up to $2000 US dollars) may discourage or prevent many strength and conditioning coaches from incorporating them into training. 

Recently, the Open Barbell^TM^ (OB) was introduced as an accessible, cost-effective (~$250 US dollars) option for measuring barbell velocity. However, establishing agreement between commercially available devices is important to determine whether such tools yield similar velocity measures for a given exercise [11]. This information allows strength coaches and researchers to confidently assess movement velocity with an understanding of the extent to which the devices in question can be used interchangeably. A recent study [12] has compared OB and TWA during the back squat (BS) exercise at 70% of 1RM. However, the purpose of this study was to determine the agreement between the OB and the TWA during the BS and the front squat (FS) exercises throughout the loading spectrum, (i.e., 30 to 90% 1RM). It was hypothesized that both linear position transducers would produce similar peak and mean velocity measurements throughout the loading spectrum. 

## 2. Materials and Methods

Participants reported to the laboratory on four separate occasions. During the first two visits, 1RM strength was assessed for the barbell BS and FS exercises, respectively. On the third visit, participants performed one set of two to three repetitions at four different percentages of 1RM in ascending order, (i.e., 30, 50, 70, and 90% 1RM) for either the BS or FS; the alternate exercise was completed on the fourth visit. During each repetition, barbell velocity was simultaneously recorded by the OB (Squats & Science, Brooklyn, NY, USA) and TWA (Tendo Sports Machines, Trencin, Slovak Republic) devices. Peak and mean velocity measurements were obtained and used to determine the agreement between devices.

### 2.1. Subjects

Eleven NCAA Division I baseball position players (19.4 ± 1.0 y; 182.4 ± 6.5 cm; 87.2 ± 7.4 kg) volunteered to participate in this research study. All participants had previous resistance training experience (5.3 ± 2.2 years) and demonstrated competency in both the BS and FS exercises. All participants completed a health and activity questionnaire to ensure that they did not have any physical limitations and were in good health. Prior to participation, all participants also completed an informed consent outlining the procedures, risk and benefits present in this study. Hofstra University Institutional Review Board approved all testing and methods prior to the participant recruitment.

### 2.2. Maximal Strength Testing

Maximal strength testing for the BS and FS exercise was performed during two separate sessions as previously described [13]. Briefly, following a standardized mobility and dynamic warm up, participants performed the BS or FS for two warm up sets at approximately 40% and 60% of their projected 1RM. Subsequently, four progressively heavier sets of two repetitions were performed with the final set being used to determine the participant’s 2RM. The 1RM was then calculated using the Brzycki formula [14]. For the FS exercise, participants were permitted to use their normally accustomed rack position, (e.g., Olympic style, arms crossed, or holding onto straps attached to the bar), and positioning was kept consistent throughout all intensity loads. Participants were also given the option to wear a weightlifting belt during 2RM attempts. All testing and trials were monitored by a certified strength and conditioning specialist (CSCS) to ensure the proper form and depth was achieved, (i.e., a thigh parallel position defined by the trochanter head of the femur reaching the same horizontal plane as the superior border of the patella) [15].

### 2.3. Barbell Velocity Testing

Participants reported to the laboratory on two separate occasions for BS and FS velocity testing. The two experimental testing conditions were randomly ordered and separated by at least seven days. During each testing session, participants performed a standardized mobility and dynamic warm up, followed by a warm up set of 6–10 repetitions at approximately 40% of their squat 1RM. The participants then performed three repetitions of either the BS or FS at 30%, 50%, and 70% 1RM in ascending order. Due to the nature of the heavy loading, participants were then asked to perform two repetitions at 90% 1RM. Each set was separated by a minimum of two minutes of rest. Participants were instructed to perform each repetition as fast as possible in a controlled manner. The velocity devices were secured to the floor and the cords of each unit were attached to the right side of the barbell just inside the ‘sleeve’ using a magnet and Velcro strap, respectively. Both devices were placed in close proximity directly below the barbell for each participant’s starting squat position. Peak and mean barbell velocity were recorded from both devices on each repetition and used for subsequent analysis. Both devices sample data of displacement during the concentric phase and calculate mean velocity as: Displacement/time. The OB calculates velocity every 2.8 mm [16], whereas the TWA incorporates an adjustable filter so that the user can adjust the minimum detectable movement (0–150 cm) [17]. The value of the filter was set to 35 cm as recommended for the squat exercise [17]. 

### 2.4. Statistical Analysis 

To determine the agreement between OB and TWA for measuring barbell velocity, the average value of peak and mean velocity recorded on each repetition performed within the same intensity load was calculated. These data, as well as differences between OB and TWA, were verified to possess a normal distribution via the Shapiro-Wilk statistic. A 2 × 4 (Device [OB vs. TWA] × Intensity [30–90% 1RM]) analysis of variance (ANOVA) with repeated measures was used to determine whether differences existed between each device’s reported peak and mean velocity scores at each intensity load. The Greenhouse-Geisser adjustment to degrees of freedom was used when the assumption of sphericity was not met. Following any significant main effect or interaction, pairwise comparisons were made between intensities using the Bonferroni adjustment to alpha, along with separate paired-samples *t*-tests between devices at each intensity load. The agreement between devices was assessed using Bland and Altman’s 95% limits of agreement (LoA) technique (bias ± [1.96 × SD of differences]) and expressed in the units of the velocity measures. Additionally, coefficients of variation (CV; antilog of SD of differences) [18] and Pearson correlation coefficients (*r*) between the average and difference of peak and mean velocity scores at each intensity load were calculated to further assess agreement. The strength of the correlations was interpreted using the following criteria: Trivial (<0.10), small (0.10–0.29), moderate (0.30–0.49), high (0.50–0.69), very high (0.70–0.90), or practically perfect (>0.90) [19]. Agreement between devices was assessed by calculating intraclass correlation coefficients (ICC_2,k_) and standard error of the measurement (SEM). Alpha was set at 0.05. All statistical calculations were performed in Excel (v. 2016, Microsoft Inc., Redmond, WA, USA). 

## 3. Results

Repeated measures ANOVA revealed significant main effects (*p* ≤ 0.005) for device and intensity for peak and mean barbell velocity in the BS and FS exercises. Regardless of device, peak and mean barbell velocity became slower (*p* < 0.001) as intensity load increased for each exercise. Across all intensities, OB reported greater peak velocity for the BS (mean difference = 0.11 ± 0.01 m·s^−1^, 95% C.I. = 0.09–0.12 m·s^−1^) and FS (mean difference = 0.11 ± 0.01 m·s^−1^, 95% C.I. = 0.10–0.12 m·s^−1^), as well as the mean velocity for the BS (mean difference = 0.01 ± 0.01 m·s^–1^, 95% C.I. = 0.00–0.01 m·s^−1^) and FS (mean difference = 0.01 ± 0.01 m·s^−1^, 95% C.I. = 0.01–0.02 m·s^−1^). Additionally, significant device × intensity interactions were observed for peak velocity recorded for the BS (F = 68.1, *p* < 0.001) and FS (F = 57.5, *p* < 0.001) exercises, where differences between devices were seen at each intensity. No interaction was found for the mean barbell velocity. Peak and mean barbell velocity, reported from OB and TWA, for the BS and FS across each intensity load are illustrated in Figure 1. 

Figure 2, Figure 3, Figure 4 and Figure 5 illustrate the Bland-Altman plots for peak and mean barbell velocity at each intensity load for the BS and FS exercises. For the BS, relationships between the differences and averages of OB and TWA peak barbell velocity scores decreased in strength as the intensity load increased (30% 1RM: r = 0.95; 50% 1RM: r = 0.73; 70% 1RM: r = 0.87; and 90% 1RM: r = 0.58). A similar pattern was observed for peak barbell velocity in the FS exercise (30% 1RM: r = 0.86; 50% 1RM: r = 0.76; 70% 1RM: r = 0.84; and 90% 1RM: r = 0.49). These relationships suggest that greater differences between the OB and TWA peak barbell velocity occurred as the average (of both devices) barbell velocity increased within a specific intensity load, and that the strength of the effect appears to decrease at greater intensity loads. In contrast, the strength of these relationships for the mean barbell velocity ranged from trivial-to-moderate for the BS (30% 1RM: r = 0.27; 50% 1RM: r = 0.35; 70% 1RM: r = 0.03; and 90% 1RM: r = 0.25) and varied between small and high for the FS (30% 1RM: r = 0.44; 50% 1RM: r = −0.27; 70% 1RM: r = 0.29; and 90% 1RM: r = 0.60), and did not display a discernable pattern. Nevertheless, CV’s were between 2% and 4% for peak barbell velocity (both exercises) and between 1% and 2% for mean barbell velocity.

The ICC’s calculated between OB and TWA suggest consistency between the devices for peak and mean velocity scores measured during both the BS and FS exercises. However, the ICC’s for mean barbell velocity (ICC_2, k_ = 0.974–0.998) were slightly greater than those for peak barbell velocity (ICC_2,k_ = 0.790–0.979). Additionally, smaller SEM’s were calculated for mean barbell velocity (SEM_2,k_ = 0.007–0.015 m·s^−1^) compared to peak barbell velocity (SEM_2,k_ = 0.040–0.119 m·s^−1^).

## 4. Discussion

The OB has been introduced as an accessible, cost-effective, and portable device for monitoring barbell velocity. However, establishing the agreement with other validated devices such as TWA is crucial prior to implementation into the strength and conditioning setting. The objective of this study was to compare the barbell velocity measured by the OB and TWA during the BS and FS exercises throughout the loading spectrum, (i.e., 30% to 90% 1RM). Our data suggest that OB consistently reported greater peak and mean velocity scores than TWA for both exercises. The differences seen in peak barbell velocity appeared to become more substantial as the average velocity, (i.e., average of both devices) increased during submaximal loads, (i.e., 30–70% 1RM) and to a lesser degree when the intensity load was increased to 90% of 1RM. In contrast, the differences seen in mean barbell velocity were generally consistent, regardless of average velocity and intensity load. The only exception occurred during FS at 90% of 1RM where differences between devices became greater as the average velocity increased. Our findings expand on a previous study that compared OB and TWA at a single intensity load [12] by comparing velocity scores at multiple intensity loads and exercises.

The use of force platforms and motion capture apparatus is generally considered the criterion method for assessing velocity-based muscle function [12,20]. However, these methods require expensive equipment, a more-detailed set-up, and their analyses are not time efficient, which makes them impractical for strength coaches and athletes. In contrast, more affordable commercially available devices, (e.g., OB, TWA, GymAware, T-Force, and FitroDyne) offer portable and practical options for strength and conditioning settings [7,9,21]. Linear position transducers monitor time and positional changes in the length of a cord attached to a person or barbell to measure velocity and calculate power based on manually-entered force, (i.e., mass of the person or barbell) data. Although their technology is simpler and less exact than force platforms and motion capture, these devices have been demonstrated to be valid and reliable for monitoring mean and peak velocity during resistance training and plyometric exercises, (i.e., BS, bench press, and counter-movement jump squats) [7,9,21,22,23,24]. Regarding the TWA specifically, Lorenzetti et al. [7] reported significant correlations for mean (*r* = 0.963) and peak (*r* = 0.932) velocity during the BS exercise at 70% of 1RM when compared to a criterion device (Vicon motion capture system). Likewise, Goldsmith and colleagues [12] reported good validity (as indicated by high ICCs, low mean bias, and narrow LoAs) of TWA and OB compared to a criterion device (Optotrack Certus 3-D Motion Capture System). 

Differences have previously been noted between various linear position transducers in their ability to measure velocity [12,24,25]. Although similar in concept, each device differs in its design and function, which impacts its estimation of velocity. For instance, OB and TWA differ in their sensitivity for detecting movement. Theoretically, the greater precision of OB also makes it more sensitive to irrelevant movements, (e.g., un-racking the barbell, brief positioning movements) that could produce very high peak velocity scores. This notion is supported by our data, as well as by evidence reported by Goldsmith et al. [12]. In both studies, peak velocity scores were markedly and systematically greater in the OB compared to TWA, and particularly more apparent at greater velocities during submaximal loads, (i.e., 30–70% 1RM). Similarly, Goldsmith et al. [12] did not report a significant difference between devices for mean velocity measures. Nevertheless, Goldsmith et al. [12] also reported greater validity, along with less error and variability, for OB as compared to TWA when compared to a criterion device (Optotrack Certus 3-D Motion Capture System). For mean velocity measures, the authors concluded that both the OB and TWA devices provide valid measurements, however some metrics suggested more accurate scores with OB compared to TWA [12]. For peak velocity measures, the authors concluded that neither device was particularly accurate, however OB appeared to be more accurate than TWA [12]. 

The results of the current study add to the literature by comparing TWA and OB throughout the loading spectrum, (i.e., 30 to 90% 1RM). The limitations of the current study include the small sample size and the use of a predicted 1RM (determined from each participant’s 2RM). The findings are also limited because OB and TWA were not compared to a true criterion. However, our findings suggest peak and mean velocity estimates to be consistent between devices for both the BS and FS exercises (as indicated by high ICCs and low CV’s). However, the systematically greater velocity scores reported by OB, particularly at greater peak velocities, indicate that these two devices are not interchangeable. Strength coaches and athletes should take these differences into consideration when comparing results obtained from either device. As both devices have been validated against the established criterion [12], it may be prudent to only use one of these devices in practice or limit comparisons to mean velocity scores.

## Figures and Tables

**Figure 1 sports-07-00125-f001:**
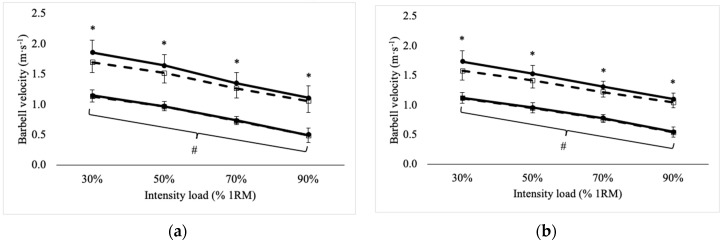
Peak and mean barbell velocity across intensity load during the (**a**) back squat and (**b**) front squat exercises. * = Significant (*p* < 0.05) difference between open barbell (OB) (Solid line with black circle) and Tendo weightlifting analyzer (TWA) (Dashed line with open square). # = Significant (*p* < 0.05) difference across all intensities.

**Figure 2 sports-07-00125-f002:**
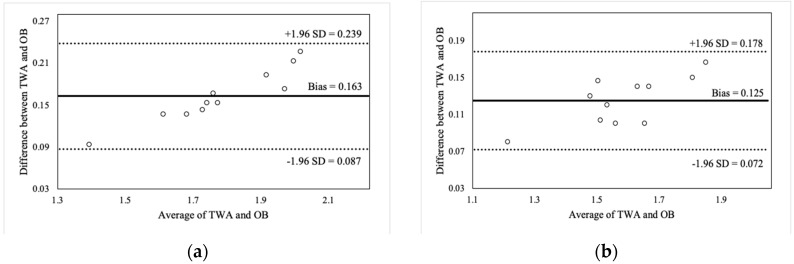
Bland-Altman plot comparisons for peak barbell velocity during the back squat at (**a**) 30% of 1RM; (**b**) 50% of 1RM; (**c**) 70% of 1RM; and (**d**) 90% of 1RM.

**Figure 3 sports-07-00125-f003:**
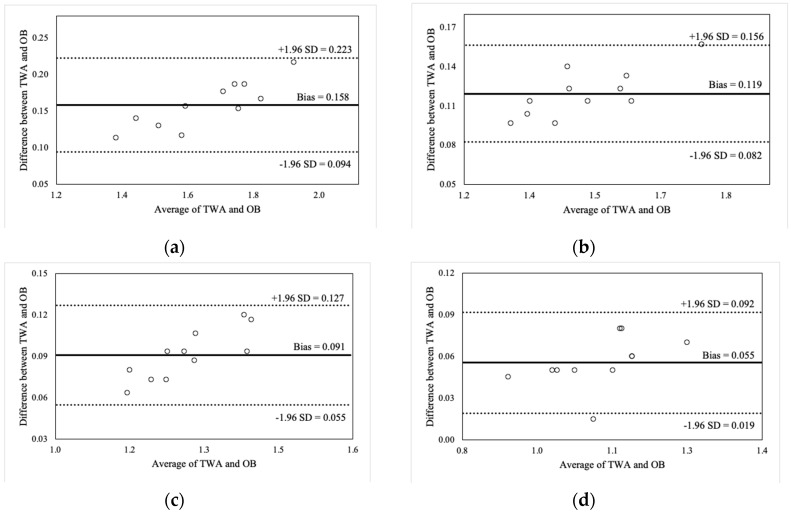
Bland-Altman plot comparisons for peak barbell velocity during the front squat at (**a**) 30% of 1RM; (**b**) 50% of 1RM; (**c**) 70% of 1RM; and (**d**) 90% of 1RM.

**Figure 4 sports-07-00125-f004:**
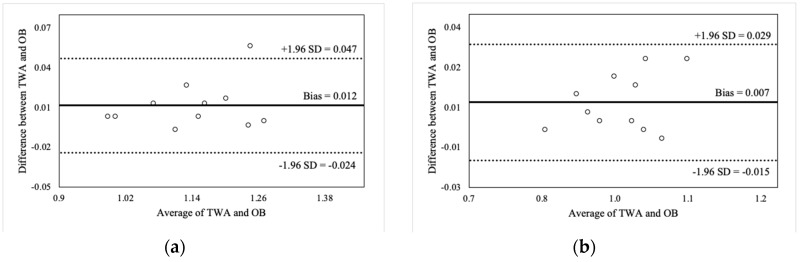
Bland-Altman plot comparisons for mean barbell velocity during the back squat at (**a**) 30% of 1RM; (**b**) 50% of 1RM; (**c**) 70% of 1RM; and (**d**) 90% of 1RM.

**Figure 5 sports-07-00125-f005:**
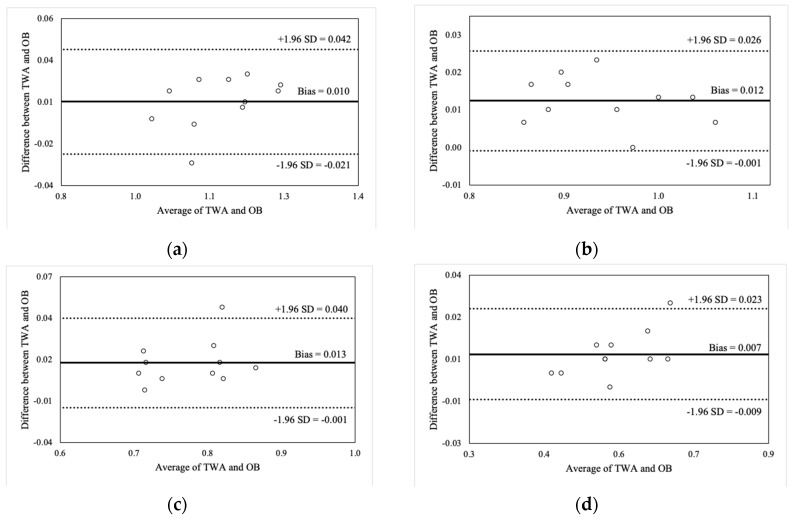
Bland-Altman plot comparisons for mean barbell velocity during the front squat at (**a**) 30% of 1RM; (**b**) 50% of 1RM; (**c**) 70% of 1RM; and (**d**) 90% of 1RM.

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
