# Peer review of "Agreement between the Open Barbell and Tendo Linear Position Transducers for Monitoring Barbell Velocity during Resistance Exercise"

_sports, 2019, doi:10.3390/sports7050125_

Round 1
Reviewer 1 Report
Authors aimed to examine the agreement between 2 commercial devices (high-cost vs. low-cost). This is clearly an interesting topic as the cost is something to be considered for strength testing and training. The article is very well written and clear for the reader. The used sample and methods are adequate for the purpose. There are just 2 small amendments I would suggest.
First, the introduction is very clear and concise. However, in the introduction authors should state that a previous studied (Goldsmith et al., 2019 IJSPP; ref [18]) as already examined this agreement and what this study adds to it.
Secondly, authors should clearly state the limitations of the study.
Author Response
Point-by-point response to reviewer 1
We would like to thank the reviewers for taking the time to carefully review our manuscript. We have responded to the concerns below and revised the manuscript as indicated with underlining. We believe the reviewers’ suggestions have greatly improved the quality of the manuscript.
Authors aimed to examine the agreement between 2 commercial devices (high-cost vs. low-cost). This is clearly an interesting topic as the cost is something to be considered for strength testing and training. The article is very well written and clear for the reader. The used sample and methods are adequate for the purpose. There are just 2 small amendments I would suggest.
First, the introduction is very clear and concise. However, in the introduction authors should state that a previous studied (Goldsmith et al., 2019 IJSPP; ref [18]) as already examined this agreement and what this study adds to it.
Response:Thank you for raising this point. We have added to the Introduction to acknowledge the previous study (Goldsmith et al., 2019). “A recent study [12] has compared OB and TWA during the back squat (BS) exercise at 70% of 1RM. However, the purpose of this study was to determine the agreement between the OB and the TWA during the BS and the front squat (FS) exercises throughout the loading spectrum (i.e., 30 to 90% 1RM).”
Secondly, authors should clearly state the limitations of the study.
Response:We have amended to clearly state and expand on our study limitations in the Discussion. “The limitations of the current study include the small sample size and the use of a predicted 1RM (determined from each participant’s 2RM). The findings are also limited because OB and TWA were not compared to a true criterion.”
Reviewer 2 Report
General Comments:
Very well done paper and I commend the authors for their clarity in writing style, clean methodology, and appropriate statistical analyses.
Methods
Line 111: Small critique, but for the sake of being consistent with physics terminology, would recommend using ‘displacement’ instead of ‘distance’.
Results
Figure 1: Though not a necessity, recommend plotting peak and mean velocities on separate figures to allow the reader to see the differences (though smaller in absolute terms) in mean velocity for the front squat and back squat. I realize that this adds figures to a paper that already has plenty, but a consideration worth mentioning given the lack of granularity in the current figure with regard to mean velocity.
Author Response
Point-by-point response to reviewer 2
We would like to thank the reviewers for taking the time to carefully review our manuscript. We have responded to the concerns below and revised the manuscript as indicated with underlining. We believe the reviewers’ suggestions have greatly improved the quality of the manuscript.
General Comments:
Very well done paper and I commend the authors for their clarity in writing style, clean methodology, and appropriate statistical analyses.
Methods
Line 111: Small critique, but for the sake of being consistent with physics terminology, would recommend using ‘displacement’ instead of ‘distance’.
Response: Thank you for this suggestion. We have made the correction.
Results
Figure 1: Though not a necessity, recommend plotting peak and mean velocities on separate figures to allow the reader to see the differences (though smaller in absolute terms) in mean velocity for the front squat and back squat. I realize that this adds figures to a paper that already has plenty, but a consideration worth mentioning given the lack of granularity in the current figure with regard to mean velocity.
Response: Thank you for this suggestion. Even when plotting mean velocity separately, there is considerable overlap and similar lack of granularity between the lines representative of OB and TWA (given that the averages were quite similar). For this reason, we would like to keep Figure 1. We also believe having mean and peak values in the same figure allows readers to clearly compare mean and peak values visually. Lastly, as the reviewer mentions, it helps reduce the already high number of figures in this manuscript.